# Nothing Like Living with a Family: A Qualitative Study of Subjective Well-Being and its Determinants among Migrant and Local Elderly in Dongguan, China

**DOI:** 10.3390/ijerph16234874

**Published:** 2019-12-03

**Authors:** Yuxi Liu, Rassamee Sangthong, Thammasin Ingviya, Chonghua Wan

**Affiliations:** 1School of Humanities and Management, Guangdong Medical University, Songshan Lake District, Dongguan 523808, China; yuxiliu123@126.com; 2Epidemiology Unit, Faculty of Medicine, Prince of Songkla University, Hat Yai, Songkhla 90110, Thailand; 3Department of Family and Preventive Medicine, Faculty of Medicine, Prince of Songkla University, Hat Yai, Songkhla 90110, Thailand; thammasin.i@psu.ac.th; 4Research Center for Quality of Life and Applied Psychology, Guangdong Medical University, Songshan Lake District, Dongguan 523808, China

**Keywords:** subjective well-being, elderly migrants, local elderly, urbanization

## Abstract

Chinese economic development has led to a significant rise in internal migration over the last 20 years, including large numbers of elderly. When elderly Chinese people migrate, they still register their residency to their place of origin and often do not register with the new administrative office at the destination due to the household registration (hukou) system in China. Thus, most of these migrant elderly do not receive full social services, possibly leading to poor subjective well-being. This study aims to qualitatively examine the level of subjective well-being and its determinants among migrants and local elderly in Dongguan City of Guangdong province, one of the most rapid economically developing areas in China. We also present the results of in-depth interviews among 27 elderly, 15 elderly migrants and 12 local elderly living in Dongguan. The results reveal that the overall subjective well-being of the two groups were good. Most migrants believed their well-being had remained stable or even improved over time due to family reunion and a better physical environment. Elderly’s most valuable needs and the main reason of migration is family reunion; however, inadequacy of social support, including community support and government support (e.g., gift during holiday season, free health examination, healthcare expenditure reimbursement), cannot be neglected for maintaining a good level of well-being. The well-being of migrant elderly can be further enhanced by promoting social services and welfare, recreational activities, and enhancing healthcare reimbursement in their new home.

## 1. Introduction

Approximately 18% of the Chinese population have internally migrated to industrialized areas during the country’s rapid economic growth in the past decade [1]. Internal migration in China, mainly involves rural-to-urban migration, identifying as ‘interregional migration’. Most internal migrants still register their residency to their place of origin and are viewed as a ‘floating population’ [2]. About 7.2% (18 million) of these were elderly (those aged 60 years or more) who accompanied their only child, primarily to take care of their grandchildren and reunite with their family [1]. Although most internal migrants look for more employment opportunities, their social services and welfare may be limited, especially in highly developed regions such as Beijing, Shanghai, and Guangdong [2]. In the past, internal migration was tightly controlled by the central government, and only in the past few decades have these restrictions been loosened. Migration reforms vary between Chinese cities in terms of their different standards for granting registered migrant households. The more developed cities usually set higher eligibility criteria (e.g., education and job responsibilities) than less developed cites for households to be registered in order for residents to receive benefits from social services and welfare including healthcare insurance. Less developed cities are more willing to attract migrants to fuel economic growth, by relaxing hukou’s constraints. However, developed cities like Beijing and Shanghai set constraints aiming to attract wealthy and highly educated migrants rather than the vast majority of rural dwellers and elderly who are unlikely to contribute to the economic growth of the region [3,4,5]. Previous studies by Liu et al. and Gui et al. found that most Chinese migrants have a lower level of subjective well-being (SWB) than local residents in host cities [6,7]. International literature on migrants’ SWB in Germany and Thailand led to the same conclusion that migrants have a low level of SWB due to their difficulty in adapting to the host city and relieving pressure [8,9]. Another stream of literature has argued migration is associated with an increase in social status and social well-being [10,11]. Most studies on international migration have shown different patterns and determinants. Studies on elderly migrants in Western countries have found that the retired elderly generally move from large cities to smaller cities with lower living costs. They conceptualized this as a form of ‘lifestyle migration’ as the result of a search for ‘self-fulfillment’ [12,13]. Studies on migrants in Middle Eastern and North Africa countries have found that young people from non-wealthy backgrounds and unemployed are more likely to emigrate, and self-select along the cultural traits of religiosity and gender-egalitarian [14,15]. 

Most previous studies on subjective well-being of Chinese people have focused on migrant workers only, neglecting migrant elderly whose social interactions and emotional status more vulnerable [16,17,18]. As their health status and financial resources decline, migrant elderly may become more disadvantaged and face more difficulties than young adults in adapting to a new environment with limited social support [1,6]. Social networks have a strong impact on well-being of the elderly. For example, interaction with family, friends, and neighbors and their support improved their life satisfaction [19]. As immigrants continue to make up an increasing percentage of the Chinese population, it is essential to understand not only to what extent migration affects their well-being changes but also the factors that contribute to well-being change. Of equal importance is the need to take into account immigrants’ perceptions of the key factors that shape a decline in well-being. In China, not only do migrant elderly experience substantial changes, local elderly also experience similar forms of urbanization and industrialization. These changes should substantially affect their well-being and therefore should also be investigated [6]. Previous qualitative studies only focused on either local or migrant elderly’s thoughts and feelings as psychological outcomes, rather than the perceptions of both locals and migrants in relation to social well-being [20,21].

This study aimed to qualitatively examine the level of subjective well-being and its determinants among migrant and local elderly in one of the most rapid economically growing areas in China. The findings should give some insights into the current well-being of both migrant and local elderly in light of China’s rapidly aging population and suggest ways to support them appropriately in order to ensure a healthy aging policy.

## 2. Materials and Methods

### 2.1. Study Area and Study Design

Dongguan City has become one of the biggest industrialized areas in China during the past decade. In 2016, 75% of the 8.3 million population immigrated there for employment or family reunion [22]. The study was conducted in collaboration with the Dongguan community committee in the areas with the highest density of immigrants, an industrial city in the Pearl River Delta and the Guangdong-Hong Kong-Macau Greater Bay.

In-depth interviews were conducted among both migrant and local elderly during December 2018 to February 2019.

### 2.2. Interview Procedure and Data Analysis

The elderly in this study were defined as any person aged 60 years or more. A migrant was defined as a person who had moved to Dongguan and had been living there for six months or more and whose household was not currently registered whereas a local resident was defined as a person whose household was currently registered in the study area. The community committee provided a list of immigrants and local elderly eligible for the study.

Initially, we purposively selected participants of diverse gender, socioeconomic status, working status, and duration of stay from the list. Using these “seeds,” a snowballing technique was then used to recruit subsequent participants until the data were saturated. Data was collected through individual semi-structured interviews. Each interview lasted for between 30 and 40 min. Interviews were audio-recorded after obtaining permission from each participant. Observational notes were also taken on nonverbal behaviors such as their facial expressions and pose/posture during the interview. Participants were asked how they perceived and defined subjective well-being (using a 5-item Likert scale ranging from very good to very poor), how it had changed compared to before they migrated (better, same, worse), and factors that qualitatively determined their SWB. Social relationships were defined as friend/neighborhood relationships and having leisure activities with friends. We also examined support given and services provided by their community and the government. We explored, in depth, the ways migrant elderly have adapted their lives to a new and changing environment. Interviews were conducted until the data were saturated. 

The data were transcribed, and thematic analysis was done. NVivo software package was used for data management and thematic coding. 

### 2.3. Ethical Considerations

Ethical approval was received from the Institutional Ethics Committee of Guangdong Medical University, China (REC: PJ2018037) and the Institutional Ethics Committee of the Faculty of Medicine, Prince of Songkla University, Thailand (REC: 61-232-18-1). Privacy and data confidentiality were ensured. Voluntary participation and unconditional withdrawal were offered to all participants. A small incentive was given as a thank you for their participation.

## 3. Results 

Out of 28 elderly invited, 27 (15 migrant and 12 local elderly) agreed to participate in the study (96.4% response rate). 

### 3.1. Baseline Characteristics of the Elderly

Of the 15 migrant elderly, all had moved from other areas in mainland China, mostly from underdeveloped and agricultural areas. The 12 local elderly had been living in Dongguan since birth. Table 1 shows baseline characteristics of the two groups. Gender, age, marital status, and working status of the two groups were similar. Most of them were aged under 70 years, married, and were not currently employed. While all migrants lived with other family members, only seven of the 12 local elderly did so. Pension was the main source of financial income, and less than half either had a salary from work or relied on interest from their own savings. Migrant elderly had some financial support from their family members, whereas the local elderly did not, regardless of whether or not they lived with other family members. The median duration of living in Dongguan for the migrant group was seven years (range: 6 months to 30 years).

### 3.2. Subjective Well-Being (SWB) Status and Its Change

Most participants acknowledged that SWB was part of life’s satisfaction; others perceived it as consisting of positive and negative emotions, moods, or feelings. Table 2 shows a summary of the self-rated subjective well-being for both groups. Most reported that they had a fair to a very good level of SWB and only one migrant, who had recently immigrated, reported having a poor level. All six migrants who had been living in Dongguan for longer than 10 years reported having a better SWB than what they experienced in their original hometown but two of the five migrants who had lived in Dongguan for 10 years or less reported having the same SWB and one reported that their SWB was worse. Three of the four migrants who had recently immigrated expressed a similar SWB and one said that it was worse. None of the migrant elderly expressed a desire to return to their hometown as long as their child still worked and lived in Dongguan. All mentioned that Dongguan was a better place for living than their hometown.

### 3.3. Perceived Contributions of SWB

Both migrant and local elderly revealed four common contributing factors to their SWB, including physical and mental health status, family relationship, social relations, and residential environment. 

#### 3.3.1. Physical and Mental Health Status

Table 3 shows a selection of statements given by the participants describing their perceptions of how health and subjective well-being are related. All elderly affirmed that physical health status was a supreme contribution to their SWB. Health was perceived as a form of capital to enable a better life. Better health conditions can lead to a better sense of happiness. Not only good physical health, but also good mental health indicated good SWB. They tried to maintain physical exercise, a positive attitude, and good mental health. Illness, on the other hand, was an important threat to SWB.

#### 3.3.2. Relationship and Harmony in a Family

The elderly moved to Dongguan mostly to be reunited with their families. Some anticipated that they would take care of their grandchildren; others wanted their children to take care of them. Keeping family harmony, in some cases, was more important than their own needs. Lack of family quality time and having family conflicts due to personal and lifestyle compatibility caused loneliness, isolation, and decreased SWB among the elderly.


*“My children invited me to come and live in the city. To be honest, I did not want to come. They are very busy at work, go out early and return late every day, I cannot see them very often, and my relatives and friends are not around, I feel very lonely.”*
(Female, 64 years old, 2-year migrant)


*“I came here mainly to take care of my grandson because my son and daughter-in-law have to work, and it is not convenient for them to take care of their son. I feel that well-being is to bring up my grandson, and then everything will be good as long as he is fine.”*
(Male, 66 years old, 6-year migrant)


*“My son and daughter-in-law treat us very well. They can take care of me if something happens to me when I am living with them. If I am sick or feel uncomfortable, they will take care of me. My wife and son are here; I feel at home. [...] Family harmony is the most important thing to me.”*
(Male, 75 years old, 12-year migrant)


*“A harmonious family can make my life easy and happy. […] Communicate more with family members and other people, do not be too stubborn.”*
(Female, 69 years old, 1-year migrant)


*“The most important thing is family harmony. I would be happy as long as the family is harmonious. Nothing else.”*
(Female, 62 years old, local)


*“Sometimes my children and I have a little friction. There are inevitable gaps between our two generations, especially with my daughter-in-law. She is so picky, much more than the nanny, making me feel very uncomfortable.”*
(Female, 63 years old, local)

#### 3.3.3. Social Relationships and Activities

Close friends filled extended family roles and shared common interests which improved their SWB. A cohesive friendship can reinforce social norms (e.g., social activities, physical exercises, and help-seeking). Good friends relived stress and prevent loneliness and isolation in their daily life. Local elderly had better social relationships than migrants. Similarly, migrants who lived longed in Dongguan had more social relationships and, in turn, it increased their SWB.


*“I have many friends in my community. We hang out together, dance, and do Tai Chi. Sometimes we have dinner in a restaurant and travel. […] We will organize some activities if some friends ask for us. […] I very much enjoy the time with my friends which increases my well-being.”*
(Female, 69 years old, 11-year migrant)


*“I can get along well with non-locals, who are my neighborhoods. Many non-locals are better than locals, and they are quite friendly. […] We play cards sometimes and share ideas about our family relationships.”*
(Male, 61 years old, local)

Some recent migrants not only lost social relationships from their homeland connection but also had difficulty making new connections in their new community. They could make daily conversation, but only superficially. They made acquaintances with people but knew very little about them. Some of them experienced a generation gap that worsened their social connection. 


*“I do not have any friends here, because I do not have a common interest with them. [...] The locals have their network, and I cannot get in. […] When we meet, we just say hello. Sometimes I cannot understand what they say. Young people will not talk to you. So I enjoy activities alone. […] There are many relatives and friends in my hometown, and we can play chess or cards together.”*
(Female, 61 years old, 4-year migrant)

#### 3.3.4. Physical Environment of the Living Area

Table 4 shows that even living in the same area, migrants and local residents had different perceptions. All migrants perceived that the overall environment of Dongguan city was better than their hometown. Good climate, clean air, as well as conveniences in daily life, such as access to public facilities, markets and banks, increased their SWB. Most local elderly perceived that the city was good, somehow only few thought they lived in a polluted and unsafe city, had poor public transport, and poor waste management.

#### 3.3.5. Community and Government Support of SWB

Without asking a prompted question, none of the participants mentioned anything about community and government support.

Community support

Migrant and local elderly had quite opposite views about community support. The majority of locals were satisfied with social support from the community. For example, community activities, holiday gifts, recreation activities, and home visits. While only a few immigrants had similar views as the locals, the vast majority of immigrants thought they had no community support, and some perceived a high level of social disparity.


*“They regularly organize activities for the elderly, such as square dancing, hiking competitions, etc. [...] Mid-Autumn Festival and Spring Festival will give us gifts such as rice and oil, as well as some daily necessities. [...] There are also dividend benefits for us in the village. I think it is very nice.”*
(Female, 62 years old, local)


*“I think it is unfair in many respects. They discriminate against us. For example, they organized an activity last year, but the community never invited us. I hope that they can treat us better, give us more benefits, and show us that they care about us.”*
(Female, 64 years old, 26-year migrant)

Government support

Nearly all locals felt that the government was supportive of welfare benefits, pension, and health care insurance, and only a few local elderly wanted more support. The majority of migrants claimed that there was a lack of support from the government in their new home. They expected more support in terms of welfare and healthcare services. Migrants reported difficulties in receiving healthcare reimbursement and what they did receive was considered inadequate. They expected that the reimbursement process could be made wherever they lived in China.

Pension


*“What satisfies me most is that the government gives us good benefits. They give us our pension on time every month. We do not have to work anymore, to have some fun.”*
(Male, 72 years old, local)


*“I want the government to give older citizens more pension due to the cost of living nowadays is much higher than before.”*
(Male, 64 years old, local)


*“I do not think there is any special support for us. I think the government treats us differently. I think it is reasonable to give the local residents better treatment, such as social benefits and welfare. However, I think it is too much.”*
(Female, 75 years old, 20-year migrant)

Health care services and health insurance


*“I am very satisfied with the welfare I receive from the government. They have done a good job in medical treatment and insurance. I can get almost whole reimbursement. This is done well. More than 10,000 yuan in all, I have to pay only several hundred yuan. More than 90% of my medical costs can be reimbursed directly.”*
(Female, 75 years old, local)


*“The government helps us with free medical examinations 1–2 times every year, and there are also family doctors who regularly come to my home to check my blood pressure and blood glucose level. […] It is great that nowadays the medical expenditure we incur at a hospital can be reimbursed.”*
(Male, 72 years old, local)


*“The community has done a good job in health care and health examination. Community health workers often come to our home to visit us and care about us. I feel very happy. After all, some people are concerned about us, which makes me feel very safe.”*
(Female, 80 years old, local)


*“Community hospitals often lack medicine, and doctors do not show great service attitude. [...] There are not many departments; the most obvious [example] is that there is no dentistry. If we get a toothache, we must go to a large hospital to see a doctor, and it is very difficult to register, and we have to wait in a long queue.”*
(Female, 69 years old, local)


*“We cannot use medical insurance here. […] I have to pay several hundred yuan each year in my hometown, but it is not available here. I have to use my own expense. My son pays for all the expenses if I am sick.”*
(Female, 67 years old, 2-year migrant)


*“The government can improve the medical insurance system. I expected that the system from the rural areas would be used in Dongguan. If it is national insurance, I think it should be available everywhere in our country. […] we cannot buy the insurance here if you have not registered your household here. So I buy it in my hometown for safety.”*
(Male, 66 years old, 6-year migrant)


*“Even if the elderly registered their household here, medical insurance cannot be transferred. You cannot reimburse the expense of outpatient services and medicines with medical insurance here. If you are admitted to hospital, you need to pay all the expenses first, and then return to your original place of residence for reimbursement. And the reimbursement rate is relatively low, about 30% of all expenditure.”*
(Female, 80 years old, 8-year migrant)

#### 3.3.6. Life Adaption of SWB Among Migrants

Most migrant elderly adapted well to the food, climate, and lifestyle in Dongguan. Their values in life, somehow, could be different from their adult children. The principle of diligence and frugality applied to elderly migrants while hedonism was their adult child’s value in life. Some elderly could accept this difference, while others tried to tolerate it silently to avoid argument. This incompatibility with their children negatively affected the elderly’s well-being. Moreover, some migrants found temporary jobs because they felt too lonely and bored staying home alone. From their expressions and language, the migrant elderly who had a job could better adapt their lives and integrate into the community.


*“I am used to eating habits and climate. There is nothing different from my hometown because we often cook at home.”*
(Male, 75 years old, 12-year migrant)


*“Due to immigration, I have to live with my son and daughter-in-law. However, I am not used to living with them because we always have different ideas, especially in terms of consumption. My daughter-in-law bought a piece of clothing worth over a thousand yuan. I do not understand, but I cannot say anything to her.” [...]*
(Female, 63 years old, 2-year migrant)


*“It was because I was too lonely, so I decided to work. I did not want to be trapped in the house all day. [...] Although I do not have any friends, I can often chat with the students, and the students like me very much.”*
(Male, 67 years old, 2-year migrant, working in the university)

Among the 15 interviewed migrant elderly, most of them thought their pension and savings could support their living expenses. Otherwise, they could earn extra money by working.


*“I am on duty here in the evenings, [I am] not too tired, earning a little money, resting during the day, or playing mahjong (a tile-based entertainment game in China) with others. I enjoy myself every day.”*
(Female, 61 years old, 4-year migrant, working in the community)

## 4. Discussion 

Our study examined the subjective well-being of both migrant and local elderly living in one of the most rapidly developing areas of China. Both groups of elderly share the same perception of SWB as life satisfaction or the balance between positive and negative emotions which are also found in other studies [23,24,25]. However, some studies defined SWB as a dimension of quality of life, something that brings joy and meaning to life [20,26]. 

Our study found that overall SWB of both local and migrant elderly were good. Most migrant elderly believed that their SWB had remained stable or even improved over time. Older adults who had positive expectations for the future and ageing were more likely to report high levels of SWB, which is consistent with previous studies [27,28]. Optimism may tend to benefit psychological well-being [29]. Migrant subjective well-being can be strongly influenced by the process of migration, which includes changes to established social networks and sources. Though migrant elderly lost their social relationships, community support, and government support that they received in their hometown, they reported having a more positive psychological attitude for ageing than local elderly. Previous studies in developed countries and other developed areas of China found that migrant elderly had higher rates of depression and had lower levels of SWB than local elderly [6,30,31,32,33]. Studies on international elderly migrants in European countries and the United States found that the migrants felt economic disadvantages, culture pressure, and exclusion of social benefits such as healthcare and welfare when they moved to the new countries [30,31,32]. Similarly, internal elderly migrants in China who moved from rural-urban areas to developed areas (e.g., Shanghai, Guangzhou) showed worse SWB than local elderly [6,33]. These studies found that migrants were vulnerable to residential segregation and inadequate social benefits [6,33]. However, a study from Hainan province, a developing area which is a popular place for retirement, showed that migrant elderly had lower rates of depression than local elderly [34]. However, those migrants had higher incomes and had higher education levels than the local elderly. Our study found a unique result of increased SWB among migrant elderly than local elderly in a developed area of China. Possible reasons include Dongguan having a larger migrant population than local residents (75% vs. 25%), thus migrants might not feel a sense of exclusion. Besides, migrant elderly are generally younger than local elderly, which supports the “healthy migrant effect” and is commonly considered as a positive outcome among immigrants [35]. Disabled or older elderly are also less likely to migrate [35]. Additionally, better climate, and physical and social environments compared to their places of origin might be positive influencing factors on their SWB.

Table 5 suggests that SWB is a psychological outcome determined by many different factors ranging from the individual to society and the environment. The elderly with good SWB report good health and harmony in family. Our study found that health was the most important asset of the participants. Previous studies showed that good health status was one of the most important prerequisites for maintaining elders’ SWB, which would make their life better [26,36,37]. Family factors, such as family reunion, were the main migration motivation of Chinese elderly, which differs from the motivation of international migrants [21,38]. The “one-child” policy has reduced the size of Chinese families. Many Chinese people, particularly the elderly, place a high value on family, which is ingrained in their Confucian culture. Taking good care of parents and children and having good relationships with family members are considered as the source of happiness [21]. Chinese elderly’s most valuable needs and the main reason of immigration is family reunion fulfillment. This can outweigh inadequacy of social support, welfare, and health insurance. 

Regarding social support and physical environment in relation to SWB, migrant and local elderly shared mixed perceptions. On the one hand, most migrant elderly suffered from a lack of social support. Social relations is considered as an important factor in experience of well-being with elderly, affecting well-being in both positive and negative ways. Loss of social relations from friends or neighborhood restricted migrant elderly to leisure activities and social activities [39,40,41,42]. Few friends and low social cohesion resulted in feelings of loneliness and homesickness. Previous studies showed that a lack of social support from friends in the migrant’s original home town was positively associated with depression among Chinese elderly [43,44]. Local elderly perceived social relations in a positive way, by which close friendship and various activities could enhance their well-being. Our study found that migrant elderly are falling through a support gap, whereby they no longer enjoy the benefits of welfare (e.g., additional financial support, free health examination, healthcare expenditure reimbursement) from their place of origin, yet they do not fully enjoy these benefits in their new place of migration. With respect to accessing healthcare, migrants complained of difficulties in receiving healthcare reimbursement in host cities, and they required out-of-pocket payments. This barrier to access to health insurance plans would make it difficult to ensure proper health services for migrant elderly. Whereas, healthcare expenditure is mostly free of charge for local residents. A study by Zhang, based on China Health and Retirement Longitudinal Study (CHARLS), found that large gaps vary across health insurance plans, where inpatient reimbursement rates for urban residence was much higher than rural residence [45]. Migrant elderly hold negative perceptions of accessibility and quality of healthcare due to perceived discrimination. They felt unfair and unhappy when recognizing the disparities in, for example, community involvement and health services, between local elderly and themselves. In contrast, local elderly could enjoy welfare fully and benefit from government policies, which made them satisfied and enhanced their SWB. For example, they could enjoy many public facilities or resources (parks, cinemas, by bus) free. Local elderly with access to urban health insurance plans are more likely to use healthcare services [44]. This result verified the “buffer effect”—a theory that accounts for the positive implications of social support for SWB [46]. On the other hand, effects on SWB from the physical environment are also important to elderly. A previous study by Paiva et al. found that physical environment is an important factor for elderly migrants’ SWB [47]. In our study, it is interesting to note that even when living in the same area, migrants showed better perception of overall environment than locals. One possible reason might be that migrants’ perceptions of the environment were shaped by the comparison between a new and old environment. Migrant elderly are living in a new neighborhood which has a comparatively better physical environment (e.g., tropical climate, convenient facilities, and transportation) to their place of origin. Few local elderly, however, felt that their environment worsened after a large influx of migrants. 

Factors related to post-migration adaptation, such as lifestyle and economic situation were associated with migrants’ SWB. In our study, all migrants were internal migrants, therefore differences in culture and language might be relatively smaller for them than for international migrants [17,48]. Besides, most migrants voluntarily migrated to be reunited with their family and to take care of their grandchildren, thus, relationships with their family in Dongguan nourished a wish to stay rather than return to the home town. Previous studies in China found that city adaptation was positively related to SWB, which in turn was associated with length of stay in the city, job satisfaction, and social support received [49,50]. 

Our study is the first qualitative study exploring the underlying influencing factors of SWB in both migrant and local elderly residents in China. Our findings add insights into the co-determinants of SWB in migrant and local elderly groups which have not been assessed in any previous studies. Our study emphasized the positive role of physical health, family ties and social support, and physical environment in enhancing the SWB of elderly. Our findings can provide evidence to help governments design healthy aging policies to improve the SWB of both migrant and local elderly at the individual, community, and national levels.

There are a few limitations which deserve mention. First, the participants were recruited through community committees, indicating that they were well connected with their communities, thus having a positive influence on their SWB. It is possible that those who were less connected with the community (e.g., being isolated or having a disability) would be more likely to have a poor SWB. Second, we did not assess the effect of changes in the socioeconomic status of the whole family on SWB. This could be explored further in future research studies. Third, the change in SWB of migrant elderly might be subject to recall bias since, for some of them, more than 10 years had elapsed since they migrated. 

There are some policy implications drawn from the findings of this study. Our study suggests that social support strongly affects elderly migrants’ SWB. Previous studies found that elderly migrants’ restricted access to social support was detrimental to their mental health [17,40,51]. Our findings confirm this point and further suggest that migrant elderly have lower social support for their health and welfare which was detrimental to their well-being. Therefore, addressing potentially discriminatory attitudes toward migrant elderly, and providing appropriate community-based support is required for reducing perceived discrimination on the side of migrants. First, elderly migrants should be taken into account in the policy of social services, and these services should be included both migrants and local residents rather than local residence only. Second, it is advisable to enhance migrant elderly healthcare reimbursement in their new home, access to healthcare needed, taking into account in service delivery of community planning. Third, more community centers and recreational space are needed to facilitate the interaction and mutual help between migrants and local elderly.

## 5. Conclusions

Overall subjective well-being among both migrant and local elderly in Dongguan is good. Migrant elderly have moved to the industrialized areas mainly for family reunion, which is the most valuable and meaningful to them. Compared to local elderly, migrant elderly have lower social and community support for their health and welfare. Subjective well-being among migrant elderly can be further enhanced by promoting social services, welfare, and recreational activities and enhancing healthcare reimbursement in their new home. The evidence provided in this study suggests that further periodic SWB evaluation is needed to ensure elderly do not experience mental health problems. 

## Figures and Tables

**Table 1 ijerph-16-04874-t001:** Characteristics of the migrant and local elderly groups (*n* = 27).

	Migrants (*n* = 15)	Locals (*n* = 12)
Gender		
Male	7	6
Female	8	6
Age (years)		
60–69	10	8
70–79	4	3
80+	1	1
Marital status		
Married	11	8
Separated/divorced/widowed	4	4
Current employment		
Yes	3	2
No	12	10
Living arrangement		
With family	15	7
Alone	0	5
Main source of income		
Pension	8	9
Family support	4	0
Salary/savings	3	3
Duration of migration (years)		
<3	4	
3–10	5	
>10	6	

**Table 2 ijerph-16-04874-t002:** Self-rated subjective well-being (SWB) among migrant and local elderly and change in SWB among migrant elderly (*n* = 27).

		Migrant (Residential Duration)			Local
		<3 years	3–10 years	>10 years	(*n* = 12)
		(*n* = 4)	(*n* = 5)	(*n* = 6)	
SWB status					
Very good	8	0	2	4	2
Good	11	2	2	1	6
Fair	7	1	1	1	4
Poor	1	1	0	0	0
Very poor	0	0	0	0	0
Change in SWB					
Better	8	0	2	6	
Same	5	3	2	0	
Worse	2	1	1	0	

**Table 3 ijerph-16-04874-t003:** Relationship between health and SWB perceived by the participants.

Participant	Good Health, Good SWB	Poor Health, Poor SWB
Female, 65 years old, 20-years migration	*“I am very satisfied with my life because my health has improved since moving here.”*	
Male, 73 years old, 7-years migration	*“I have high blood pressure, though it can be controlled by taking medicine, yet the doctor still advised me to do some exercises, such as walking and Tai Chi. Therefore, I take a walk in the park with my husband after dinner every night and do Tai Chi in the morning at my leisure. Everything will be good if you have a good body, and my child does not have to worry about us.”*	*“If you are in poor health, not only you have to suffer, but also your family gets involved.”*
Female, 80 years old, 8-years migration	*“Good health is the most important thing. It is better that I do not get sick and do not have to be admitted to hospital. I do not ask for anything. Life is good enough.”*	
Male, 64 years old, local	*“I like to do physical exercise, which can keep me healthy; I do not like to play cards. After all, a healthy body is the base of doing everything. […] I run at five o’clock almost every day.”*	
Male, 67 years old, local	*“I think the best mentality is to keep open-minded, and then you will feel that everything around you is good. Keep open-minded and not being too hard on everyone may improve our well-being.”*	
Female, 69 years old, local		*“I feel that my body is not good, which made me distressed. I spent 2000 yuan in the hospital a few days ago to buy drugs for chronic diseases. It was painful that the money spent on medicine now exceeds the cost of eating. I have to take medicine for chronic diseases every day. Now my money is all spent on the hospital. […] It is because of insomnia that I often wake up at 2:00 or 3:00, which makes me upset.”*
Male, 71 years old, local	*“People can be happy as long as we can eat and work with a healthy body. Life would be great when we eat well and entertain ourselves well, which could be better if we live without illness and pain.” and “I want to live to a ripe old age with a healthy body as well as make some changes of my life and have a happier family.”*	

**Table 4 ijerph-16-04874-t004:** Summary of participants’ perceived quality of environment (*n* = 5).

Environment	Participant	Recodes
**Good**	Male, 71 years old, 8-year migrant	*“The air and climate here are better than in my hometown, which makes people very comfortable. I feel my body is better now and I hardly get sick. The urban environment is much better than the rural areas, it is very clean everywhere, and the greening is also very good. [...] The climate is too cold in my hometown; I do not want to go back.”*
	Male, 65 years old, 4-year migrant	*“It is very convenient to do grocery shopping here. You can go to the market or supermarket with only a few minutes’ walk. Fresh fruits and vegetables are always available. Living here is much more convenient than living in my hometown; I do not have to worry about the necessities of life. [...] It is also convenient to go to the bank, and automatic teller machines can be seen everywhere. […] We also participated in run activities in our community, and made use of public resources such as libraries and museum.”*
**Poor**	Male, 71 years old, local	*“I hope that they will improve the river. It is so smelly. It really disgusts me. The water quality in the river is very bad. The river is filled with stagnant water, the river is blocked, and the water is black as ink. […] The water quality in the river is very bad. It is not so stinky these two days because they have flowing water. If not, a terrible smell will come from it. Also, the mosquitos will breed.”*
	Female, 60 years old, local	*“I am not satisfied with the public transportation system. Because where we live is too remote. It is not convenient to take the bus. There used to be a bus from there, but it was canceled because few people took it.”*
	Male, 66 years old, local	*“There are many non-locals here; they rent houses here, creating much garbage, and the environment has become worse. Also, many rental houses will definitely cause security problems.”*

**Table 5 ijerph-16-04874-t005:** Perception of effects of determinants on SWB among migrant and local elderly.

Determinants	Perceived Effects of Determinants	
	Migrant	Local
Health status	++	++
Family relationship	++	++
Social relations	-	+
Physical environment	+	+/-
Community support	-	++
Government support	-	++
Life adaption among migrants	+/-	NA

+ good: - poor, +/- has both views, NA: not applicable.

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
