# Peer review of "Nothing Like Living with a Family: A Qualitative Study of Subjective Well-Being and its Determinants among Migrant and Local Elderly in Dongguan, China"

_ijerph, 2019, doi:10.3390/ijerph16234874_

Round 1

Reviewer 1 Report

Referee report on “Nothing like living with a family: A qualitative study of subjective well-being and its determinants among migrant and local elderly in Dongguan, China”

About the paper:

This paper aims to qualitatively examine the level of subjective well-being and its determinants among two groups - migrants and local elderly - in Dongguan City of Guangdong province. It relies on qualitative data of 27 interviews with the above-mentioned groups carried as follows: 15 elderly migrants and 12 local elderly living in Dongguan. The results show that most migrants reported similar or higher levels of social well-being due to unity with family and a better physical environment. The paper ends by suggesting a policy that enhances social services and welfare to improve the elderly’s social well-being. However, the paper needs to be improved to be published. Having said that, find below a list of suggested improvements.

Abstract: The introductory sentence on internal migration needs to be cited. The second sentence about people who migrate and do not register with the new administration needs to be clarified and cited. The statement of the inadequacy of social support is not clear/coherent. Introduction: As in the abstract, a statement on internal migration without citation. The author mentioned “higher eligibility criteria for households to be registered” but did not state what are those criteria? And why they are higher now?

I would suggest stating a comparison between old and new criteria.

A part that starts with “As their health status” until “should also be investigated” is not clear; those are results or literature? I recommend the authors to add some evidence from different regions such as the Middle East and North Africa (MENA) region SSA. For example, you can refer to some recent applications: Dibeh, G., Fakih, A., & Marrouch, W. (2018). Decision to emigrate amongst the youth in Lebanon. International Migration, 56(1), 5-22. Docquier, F., Tansel, A., & Turati, R. (2019); Do Emigrants Self-Select Along Cultural Traits?: Evidence from the MENA Countries. Evidence from the MENA Countries, Forthcoming, International Migration Review. Material and Methods: Study area and study design: The interviews were conducted among 27 elderly, which might not give a good picture of the population in Dongguan City (small/unrepresentative sample size) Interview procedure and data analysis: Is this section actually needed? Is it important for the targeted audience to tell them that the researcher: contacted the people to be interviewed? received a written informed consent? gave the participants small gifts as appreciation?

If this is being mentioned for ethical reasons, then it is recommended to move it to the “Ethical Considerations” section or to merge them all under one section called “Data and Methodology”

Results: Subjective well-being (SWB) status and its change The number of observations is 27, however, in table 2 the data on the “Change in Social Well-Being” is 15 (8+5+2) rather than 27 (as in the data on the SWB status).

This might be a result of missing observations. If so, then it is recommended to mention it.

Social relationships and activities The first sentence defines the social relationship, however, the author is discussing the results of the study. Hence, it is recommended to move any variable definition to the “Material and Methods” section. The second sentence needs more clarification, i.e. how close friends help in the improvement of the social well-being of the elderly? Conclusion: the author is advised to elaborate more on the policy implications, showing if it agrees or disagrees with previous literature that tackled this/similar issues

Author Response

Dear reviewer:

First of all, we wish to thank you for your constructive, encouraging and useful comments. We have tried our best to carefully respond to you point by point.

Kindly regards,

Yuxi Liu

Reviewer 2 Report

The article is well-written and clear; methods are clear and well-described and the discussion and conclusions are clear and can be derived from the results. It also provides some directions for policy. This is a qualitative paper, yet it misses some of the depth that especially qualitative research is suited for. The data are presented as a summary and almost in a quantitative way sometimes. I feel here a lot more can be gained. Moreover, the academic puzzle or academic relevance could be addressed more in-depth and could be stronger. These are some of my suggestions:

1) The introduction is really short and only addresses the possible societal relevance of the study. However, it nowhere refers to the larger literature and how this study feeds into that. What can we learn from this study for the wider field? What academic gaps are addressed by the study? Et cetera

2) In the abstract it reads that the results show that subjective well-being is not only a psychological outcome but is influenced by multilevel factors. But all the literature already says this. This is well-established. Hence, what is your academic contribution beyond that?

3) What is the theoretical framework or analytical framework you are employing? You could say a bit more about this in the introduction. What are the mechanisms you are trying to explain/ explore/ lay bare? It does not have to be extensive, but is needed to better interpret the findings.

4) As said above, you could with qualitative data go more in depth. Now it is more sort of a descriptive comparison, an in-depth analysis is missing. That is because no linkages are made between what certain people said and the characteristics of these people. Or what they experienced in the different themes and how that relates.

5) Moreover, you only have 15 people in one group and 12 in the other. This is fine for doing in-depth qualitative analyses in which you do the above and you discuss experiences of people in depth. However, if it is enough to do a real comparative analysis that is more descriptive (saying this many people said this and this many people said that), I wonder. Especially because of selection bias, that is acknowledged in the limitations. Participants were recruited through community committees and socioeconomic status could have played a role. Also, some migrated over 10 years ago. That is a very different story from migrating 2 years ago. They would be more likely to be integrated into the society and have less difficulties with social relationships in the neighbourhood. Finally, do you really need a 'control group' for this study? You want to show what particularities the migrant elderly have to face and how this affects their SWB. You do not need the other group to describe this and go in depth, actually sometimes it keeps you from that in-depth story.

6) Another important distinction I have observed and that might greatly affect the results is that all the migrants have migrated to live with family while about half of the locals live by themselves. These create different dynamics. So are you picking up migration effects or something else namely living together as a family. There are various studies showing how elderly of whom children have migrated experience ill well-being because there is no family to take care of them in their old age and they feel alone. This migration therefore could have logically greatly increased their health. This has not been reflected on. Neither whether of those locals who live alone, the family is living near. Maybe they can be in the situation some of the migrants where in before migration. 

7) I think that it is a relevant study, but some parts are much more relevant than others and have more of a story than others. Let me explain. That physical and mental health; family relationships; social relations and residential environments are important to both groups is logical. These are factors also found elsewhere. What is then described under the first heading does not add to the literature as you would not expect any differences between the two groups under study and it doesn't reveal anything about migration or reunification. The same goes for family: you have only described those living with their family and the experiences would be the same. You have not gone into specifics for migrant families. However, after that comes I think the more interesting part where you could go more in-depth. I think there the story comes out, so you might think of deleting or only mentioning the previous two shortly. Social relationships is where there is a difference to be expected and how that affects well-being is relevant. This especially holds for health care services and health insurance. I think here you can see that migrants have difficulties that are particular to their situation and you can explain how that affects their SWB. So e.g. on page 11 line 341-345, this is an important and relevant conclusion and this is where you can spend more time on and explore further.

Author Response

Dear reviewer,

We wish to thank you for your constructive, encouraging and useful comments. We have tried our best to carefully respond to you point by point.

Kindly regards,

Yuxi Liu

Reviewer 3 Report

I find the research question interesting and the authors did a great job collecting data on highly relevant issues in China. The Chinese society is aging and little information is known about this growing group of people in the Chinese society.

However, I am sceptical about the quality of the data. The data set only covers 27 interviews of persons living in the same area. One may ask if these data is representative. What is the value added compared to larger data sets covering more regions and more people. For instance the CHARLS data covers the same set of questions but for a much larger number of observations.

The authors should try to convince the reader about the advantages of their data/approach. Moreover, it would be nice to see comparisons between their results and results published based on the CHARLS data.

As far as I know, the CHARLS data is more standardized without giving the interviewed people the possibility to express their opinion. The interviews conducted for this study give more room for individual answers but those answers may be difficult to compare across individuals.

The authors may want to add a lengthy comparision of the two data sets.

Author Response

(The authors gave the same response as above.)

Round 2

Reviewer 2 Report

Dear authors, 

The paper has significantly improved and I see you have taken some of the suggestions of the reviewers into account. Especially the engagement with the literature has improved and this makes your paper much more academically relevant for the reader. The social networks stand out more now as a factor and that is now a much clearer story.

I below address your response to my comments and my response to that.

My comments:

1) The introduction is really short and only addresses the possible societal relevance of the study. However, it nowhere refers to the larger literature and how this study feeds into that. What can we learn from this study for the wider field? What academic gaps are addressed by the study? Et cetera

Your response: Literature on societal relevance and academic gaps have been added on page 2, lines 67-81.

My current response: This has greatly improved and you have situated your paper much better in the larger debate. Only do some copy-editing here as there are some grammatical mistakes and typos.

2) In the abstract it reads that the results show that subjective well-being is not only a psychological outcome but is influenced by multilevel factors. But all the literature already says this. This is well-established. Hence, what is your academic contribution beyond that?

Your response: We have added that these multi-level factors are inter-related and some may have more influence than the others. This depends largely on belief, culture and context (Page 1, Lines 25-27)

My current response: Yes it has improved. However, I would still argue that this has been shown by others and this is fine, but it is not your main contribution. I think you are underselling yourself, as you find it is interrelated to some specific context that is so relevant for this paper. Maybe by removing "The findings of the in-depth interviews showed that 23 subjective well-being is not only a psychological outcome but is influenced by multilevel factors, such 24 as the individual, social conditions and the local environment. These factors are inter-related and 25 some have stronger influence to subjective well-being than the others. This depends largely on belief, culture and context." And immediately going into what specifically you found for migrants and and non-migrants, we go into your contribution specifically. 

3) What is the theoretical framework or analytical framework you are employing? You could say a bit more about this in the introduction. What are the mechanisms you are trying to explain/ explore/ lay bare? It does not have to be extensive, but is needed to better interpret the findings.

Response: Theory of social networks was used to inform the more detailed analysis of subjective well-being and cited (Page 2, Lines 68-74).

My current response: Much clearer.

4) As said above, you could with qualitative data go more in depth. Now it is more sort of a descriptive comparison, an in-depth analysis is missing. That is because no linkages are made between what certain people said and the characteristics of these people. Or what they experienced in the different themes and how that relates.

Response: We narratively described our findings in the result part and discussed in great details in the discussion. For example, how related factors were influenced subjective well-being was shown in Table 5 (Page 11-12, Lines 374-375).

My current response: Maybe it is a disciplinary thing (coming from Cultural Anthropology and Sociology), but I still miss a bit more in-depth analysis or material. Table 5 does show how it interrelates, but it could be discussed more in-depth in the text. Although there is now little room for it.

5) Moreover, you only have 15 people in one group and 12 in the other. This is fine for doing in-depth qualitative analyses in which you do the above and you discuss experiences of people in depth. However, if it is enough to do a real comparative analysis that is more descriptive (saying this many people said this and this many people said that), I wonder. Especially because of selection bias, that is acknowledged in the limitations. Participants were recruited through community committees and socioeconomic status could have played a role. Also, some migrated over 10 years ago. That is a very different story from migrating 2 years ago. They would be more likely to be integrated into the society and have less difficulties with social relationships in the neighborhood. Finally, do you really need a 'control group' for this study? You want to show what particularities the migrant elderly have to face and how this affects their SWB. You do not need the other group to describe this and go in depth, actually sometimes it keeps you from that in-depth story.

Response: A total of 27 elderly in-depth interview could give us insightful and understanding about subjective well-being among Chinese migrant elderly as elaborated in the result part.

Selection bias in our study may exist but considered to be low. Majority of migrants were known by the community committee, because they had to register to get a residence permit. Our study was also designed to minimize this bias by recruiting elderly with diverse gender, socioeconomic status, working status and duration of stay explained on page 3, lines 103-104. Being aware of this bias nature in purposive sample helps us to cautiously interpret our results.

Duration of stay is also our interest and concern. That is why we ensured to have them varying in their duration of stay and reported the findings in the results.

We did not have a control group in our study. Our objectives were to understand subjective well-being between the two groups. The findings will provide some directions for policy on both local and migrant elderly.

My current response: With my comments I did not want to imply that you cannot do the research you do because of selection bias (and thank you for clarifying further); it was more about how you write about this comparison. And that you should be aware of it when making the comparison. What can you write about and what not. So exactly what you right in your response. Again, I think you have a great and interesting story to tell, but just be careful with some of the conclusions on what the reasons are for that difference you describe between the groups. The same holds for ‘control group’, I know you do not have that. That why I put it in ‘ ‘. It was just how you write about the two groups sometimes and how you compare and contrast that led me to use this word.

6) Another important distinction I have observed and that might greatly affect the results is that all the migrants have migrated to live with family while about half of the locals live by themselves. These create different dynamics. So are you picking up migration effects or something else namely living together as a family. There are various studies showing how elderly of whom children have migrated experience ill well-being because there is no family to take care of them in their old age and they feel alone. This migration therefore could have logically greatly increased their health. This has not been reflected on. Neither whether of those locals who live alone, the family is living near. Maybe they can be in the situation some of the migrants where in before migration. 

Response: We have reported how migration greatly affect the elderly subjective well-being. We also made a comparison among many factors related to subjective well-being in Table 5 in the discussion part.

We found that whether the locals live with alone, their situation were varying from different families. Though locals live alone, they still could see and go out with their family if they would like to. The situation was different with migrants before migration. The migrants suffered the splitting of extended family.  

My current response: My remarks are exactly what you say in the last part of your response that health for migrants improved as they joined their children and therefore could have better health than the locals who might be separated at the moment of interview. But from the text it was not clear that the locals all lived close to their families. It could have been that their children lived somewhere else. So now you have clarified it, it cannot be a factor. Thank you for your answer.

7) I think that it is a relevant study, but some parts are much more relevant than others and have more of a story than others. Let me explain. That physical and mental health; family relationships; social relations and residential environments are important to both groups is logical. These are factors also found elsewhere. What is then described under the first heading does not add to the literature as you would not expect any differences between the two groups under study and it doesn't reveal anything about migration or reunification. The same goes for family: you have only described those living with their family and the experiences would be the same. You have not gone into specifics for migrant families. However, after that comes I think the more interesting part where you could go more in-depth. I think there the story comes out, so you might think of deleting or only mentioning the previous two shortly. Social relationships is where there is a difference to be expected and how that affects well-being is relevant. This especially holds for health care services and health insurance. I think here you can see that migrants have difficulties that are particular to their situation and you can explain how that affects their SWB. So e.g. on page 11 line 341-345, this is an important and relevant conclusion and this is where you can spend more time on and explore further.

Response: Our study showed the opposite direction of what have been understood. We have explored more and added this issue in the “Discussion” of the revised manuscript. First, we have improved the social relations factor affect subjective well-being in both positive way and negative ways and cited on page 12, lines 378-380 & lines 383-385. Second, with respect to support gap for migrant elderly, we explained more detail how access to healthcare and social welfare affect their subjective well-being. We have explored more information on page 12, lines 386-395 & lines 398-400.

My current response: It is much clearer now. Also that social networks are addressed in the introduction makes it clear that is where the story and contribution is.
